# The work of return to work. Challenges of returning to work when you have chronic pain: a meta-ethnography

Mary Grant,[1] Joanne O-Beirne-Elliman,[1] Robert Froud,[1,2] Martin Underwood,[1] Kate Seers[3]

¹Clinical Trials Unit, University of Warwick, Warwick Medical School, Coventry, UK
²Department of Health Sciences, Kristiania University College, Oslo, Norway
³Warwick Research in Nursing, University of Warwick, Warwick Medical School, Coventry, UK

**Correspondence to**
Dr Mary Grant;
M.Grant.2@warwick.ac.uk

## ABSTRACT

**Aims** To understand obstacles to returning to work, as perceived by people with chronic non-malignant pain and as perceived by employers, and to develop a conceptual model.

**Design** Synthesis of qualitative research using meta-ethnography.

**Data sources** Eleven bibliographic databases from inception to April 2017 supplemented by citation tracking.

**Review methods** We used the methods of meta-ethnography. We identified concepts and conceptual categories, and developed a conceptual model and line of argument.

**Results** We included 41 studies. We identified three core categories in the conceptual model: managing pain, managing work relationships and making workplace adjustments. All were influenced by societal expectations in relation to work, self (self-belief, self-efficacy, legitimacy, autonomy and the meaning of work for the individual), health/illness/pain representations, prereturn to work support and rehabilitation, and system factors (healthcare, workplace and social security). A mismatch of expectations between the individual with pain and the workplace contributed to a feeling of being judged and difficulties asking for help. The ability to navigate obstacles and negotiate change underpinned mastering return to work despite the pain. Where this ability was not apparent, there could be a downward spiral resulting in not working.

**Conclusions** For people with chronic pain, and for their employers, navigating obstacles to return to work entails balancing the needs of (1) the person with chronic pain, (2) work colleagues and (3) the employing organisation. Managing pain, managing work relationships and making workplace adjustments appear to be central, but not straightforward, and require substantial effort to culminate in a successful return to work.

## INTRODUCTION

Chronic pain, defined as pain lasting 3 months or more,[1] is a global public health problem affecting one in ten adults.[2] A 2017 mega-ethnography brought together 11 qualitative evidence syntheses to explore the experience of living with chronic non-malignant pain.[3] Previous reviews have identified the importance of the effect of chronic pain on people's work life.[4 5] Chronic pain is strongly

## Strengths and limitations of this study

► This is the first study to present employer and employee perspectives together.
► This study draws together what is known from qualitative studies to inform practice.
► This study highlights health and illness and pain representations in relation to return to work.
► Only five studies covered employers' perspectives, so there are fewer data on employers' perspectives compared with the perspectives of people with chronic pain.

associated with claiming disability and unemployment benefit in Australia[1] and with unemployment in the USA.[6] The obstacles to staying in work for people with musculoskeletal pain have previously been explored in a meta-ethnography,[7] and factors promoting staying at work are the focus of a previous mixed-methods systematic review.[8] A qualitative systematic review of the impact of chronic pain in the workplace[9] takes a broad perspective including impact on employment status, sickness absence and loss of productivity in contrast to a condition and gender-specific literature review focused on work and rehabilitation for women with fibromyalgia.[10] There is a qualitative research on the perspective of doctors,[11] but this is not considered further in this paper.

The lack of focus on return to work for people with chronic non-malignant pain and the perspective of employers presents a knowledge gap in existing reviews. Return to work can refer to the process of returning after a period of sick leave[12] or returning after a period of unemployment.[13] This review uses qualitative evidence synthesis to increase understanding of the obstacles to return to work for people with chronic pain and their employers, and this can then inform intervention development to support return to work.[4 14]

## METHODS

### Aims and objectives

This meta-ethnography explores experiences of returning to work, as perceived by people with chronic non-malignant pain and by employers, and develops a conceptual model.

### Study design

There are two main approaches to synthesising qualitative research, one that aggregates findings to describe the literature and one that aims to interpret findings and develop a conceptual understanding.[4 14 15] Meta-ethnography is an interpretative form of knowledge synthesis that was chosen for this study in order to both integrate and develop a greater understanding of existing knowledge and identify any other overarching concepts that would explain the data. The seven phases of meta-ethnography are outlined by Noblit and Hare[15] and elaborated on by Toye *et al*.[14] These are (1) getting started by identifying the area of interest; (2) deciding what is relevant; (3) reading and rereading the studies; (4) determining how the studies are related, which involves creating a list of key phrases, ideas, metaphors and concepts; (5) translating the studies into one another, where direct comparisons are made and similar concepts are sorted into categories; (6) synthesising the translations, where researchers make sense of the conceptual categories to develop new knowledge and understanding; and (7) expressing the synthesis. A line of argument was constructed by examining how the conceptual categories relate to each other.

### Identifying and appraising the review articles

#### Search methods

##### Study selection

Eleven electronic bibliographic databases were searched (Allied and Complementary Medicine (AMED); Applied Social Science Index and Abstracts (ASSIA); The Cumulative Index to Nursing and Allied Health Literature (CINAHL); EMBASE; International Bibliography of the Social Sciences (IBSS); MEDLINE; PsycINFO; Social Services Abstracts; Sociological Abstracts; Web of Science and Westlaw) from inception up until 25 April 2017, supplemented by backward and forward citation tracking using Scopus. These databases were considered appropriate because in early scoping work we identified relevant studies in these databases. An academic support librarian undertook the initial search in collaboration with RF in December 2016, and this was updated by MG in April 2017, who continued the screening and selection of papers. The search terms used included 'Chronic pain' and 'Return to work (MeSH) OR Employment OR Employer OR Supported Employment (MeSH)'. In April 2017 two additional search terms were used, 'pain' to broaden search as 'chronic pain' was not identifying all relevant papers, and 'qualitative' as suggested by Shaw *et al*[16] to focus the search on studies with this type of methodology. The search strategy is detailed in online supplementary file 1. All qualitative studies using face-to-face interviews and focus groups which explored perceptions of obstacles to return to work, in employers and people who were off work, sick-listed and had chronic pain, were included. Non-English-language texts were excluded.

#### Quality appraisal

The quality of studies was evaluated using the Critical Appraisal Skills Programme (CASP) qualitative assessment tool.[17] A scoring system was used for CASP (yes=3, can't tell=2, no=1). A score of 20 or higher indicates the paper is deemed to be of satisfactory quality. The GRADE-CERQual (Grading of Recommendations Assessment, Development and Evaluation-Confidence in the level of Evidence from Reviews of Qualitative research) approach was also completed.[18 19] Confidence in review findings was assessed based on four components: adequacy of data,[20] coherence,[21] methodological limitations[22] and relevance.[23]

### Analysis

Initially, the first 10 papers (in alphabetical order of author) were read by MG, KS and JO-B-E in order to identify key 'concepts', the raw data of meta-ethnography.[14] These concepts are ideas drawn from the findings of the original papers. They are also known as second-order concepts because they are the authors' interpretations of the participants' narratives (known as first-order concepts).[24] The participants' narratives chosen by the author are examples of second-order concepts.[14] After reading these 10 papers, the concepts identified by each researcher were amalgamated through discussion and grouped into conceptual categories that the team then worked collaboratively to name. This took place over a series of three meetings. These conceptual categories are third-order concepts insofar as they are the researchers' interpretations of second-order concepts. All concepts were identified by all three authors (KS, MG, JO-B-E), and even if exact wording differed the concept was the same. This is the way that studies were translated and related to each other. The first author then proceeded to read the rest of the papers and continue this process of analysis. Five additional papers were also read by KS and JO-B-E where MG felt a collaborative discussion would be helpful due to the nature and/or findings of the studies. Thus 25% of papers were checked (n=10), then an additional 10% were checked (ie, 35% in total) to ensure ratings and concepts were in agreement. All the included papers were uploaded to QSR International's NVivo V.11 software,[25] and nodes were created for the conceptual categories. The next stage was to make sense of these categories through further discussion, make decisions about which were the core categories and develop a line of argument and conceptual model,[14] involving a further four meetings. Recurring and common concepts were compared across studies,[15] where directly comparable (reciprocal translation) together they contributed to our line of argument. We did not find studies that stood in opposition (refutational translation). The line of argument makes a whole of something more than a sum of the parts.[15] MG,

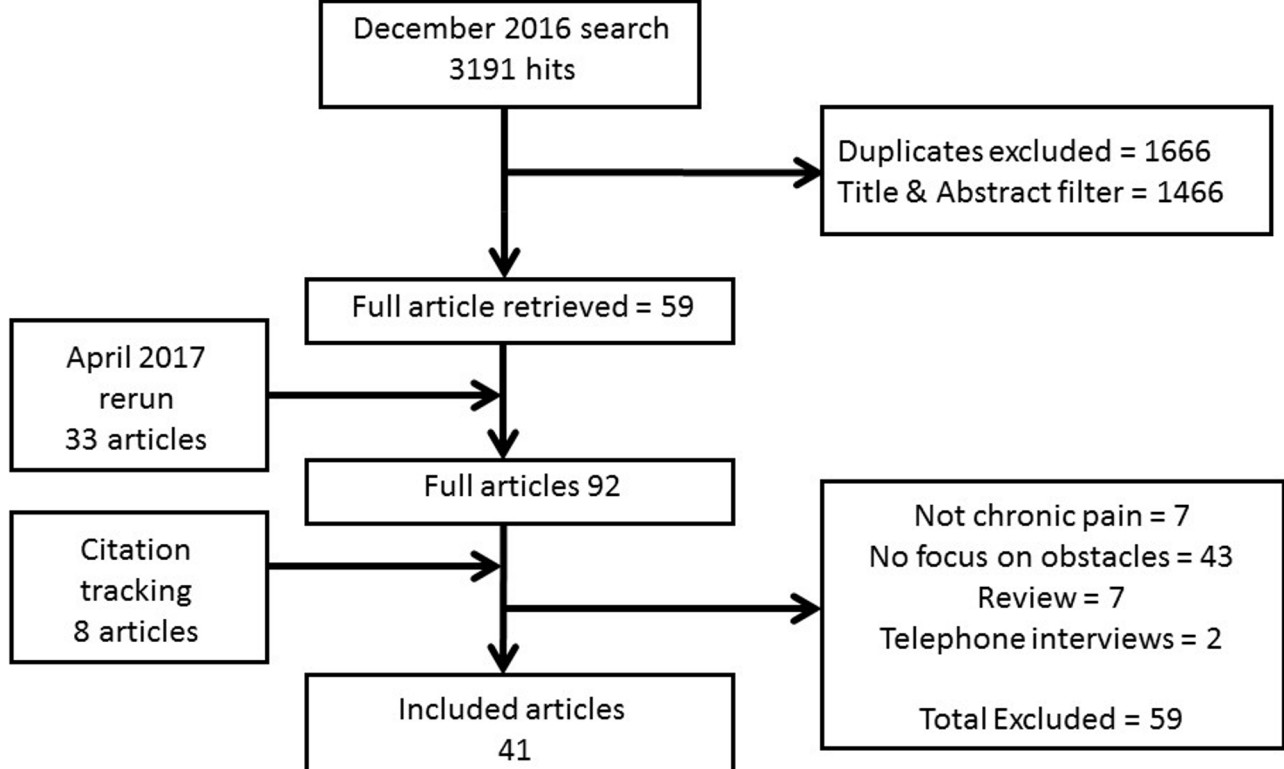

**Figure 1** Flow chart illustrating search outcome.

JO-B-E and KS independently drew their own conceptual model before coming together to agree a model, which was revised through several discussions, and the final version is presented in this paper. The culture described by Toye *et al*[26] of a core team that provided a safe environment in which to freely discuss, agree, disagree and change their position in relation to conceptual analysis was seen as a key strength, laying the foundations for a rigorous review. This approach was adopted in this review. We explored alternative interpretations and explanations, including locus of control, navigating relationships, normalising participants' pain condition and agency but these ideas were not supported as major concepts. Many of these concepts were subsumed in other categories.

### Patient and public involvement

A patient and public representative was involved in the development of the research funding submission for the overall study as a coapplicant and endorsed the importance of the focus of this meta-ethnography recognising the central nature of obstacles to return to work.

### RESULTS
### Search outcome and overview of studies reviewed

We include 41 papers and the search outcome is illustrated by a flow chart in figure 1. The initial 3191 hits were screened by titles and abstracts, duplicates excluded, and a further 1466 were excluded at this stage. Following the reading of full texts, papers were excluded as they were neither about chronic pain nor specifically about return to work. All studies that were critically appraised passed the first two screening questions of the CASP tool that related to whether there was a clear statement of the aims of the research and if qualitative methodology was considered appropriate to address the research goal.[17] CASP scores are presented in online supplementary file 2. Of the 41 articles included, 32 reported interview studies and 9 focus group studies. Twenty-one studies were from Scandinavia (14 in Sweden, 4 in Norway and 3 in Denmark), 7 were from the UK, 7 were from Canada, 2 in France, and 1 each from Australia, South Africa, Switzerland and USA. Only five studies were from the employer's perspectives. One study included in the review did not specify the type of chronic pain, but the majority of the studies involved people or employers of people with musculoskeletal pain, mainly affecting the back and neck, and some were injury/work-related. Studies of people with musculoskeletal disease, including arthritis, fibromyalgia and systemic lupus erythematosus, were also included (table 1).

### Overarching conceptual categories

A total of 342 concepts were clustered into 16 conceptual categories summarised in table 2. The first column of table 2 contains third-order concepts. We worked with second-order concepts, and the second column of table 2 is second-order data, some of which are illustrated with first-order participant quotations. This table also highlights the CERQual profile. The three key conceptual categories identified by the team are described in this section. The balancing of these three inter-related categories

**Table 1** Description of included studies

| | Author and year of publication | Country | Type of pain | Number, gender and age of participants | Participants | Methods of data collection | Methodological approach—analysis |
|---|---|---|---|---|---|---|---|
| 1 | Ahlstrom et al,[12] 2017 | Sweden | Neck pain. | 16 women, mean age of 54. | People with history of long-term sick leave in human service organisations. | Interviews. | Constructivist grounded theory approach. |
| 2 | Andersen et al,[42] 2014 | Denmark | Back or upper body. | 4 men and 3 women aged 33–57. | Participants in chronic pain self-management programme or tailored physical activity programme. | Semistructured interviews. | Systematic text condensation—thematic cross-case analysis. |
| 3 | Angel et al,[53] 2012 | Denmark | Low back pain. | 20 (65% women), mean age of 46. | Participants of counselling intervention addressing workplace barriers and physical activity. | Semistructured clinical interviews. | Narrative analysis. |
| 4 | Ashby et al,[58] 2010 | Australia | Chronic low back pain. | 11 men aged 23–59. | Participants in a work hardening programme. | Semistructured interviews (ethnographic). | Thematic content analysis. |
| 5 | Brooks et al,[48] 2013 | England | Persistent, non-specific low back pain of at least 12 weeks in duration. | 6 women and 3 men. Working participants (5) aged 45–52 (mean 49.2). Unemployed participants (4) aged 51–63 (mean 57) and their significant others. | Participants from hospital pain management clinic. | Semistructured interviews. | Template analysis style of thematic analysis. |
| 6 | Buus et al,[59] 2015 | Denmark | Low back pain. | 25 (56% women) (mean age 46.8). | People who had received counselling intervention designed to motivate them to change work routines and exercise. | Semistructured interviews. | Interpretative thematic analysis. |
| 7 | Coole et al,[45] 2010 | UK | Low back pain. | 13 women and 12 men aged 22–58 years (mean age 44.7). | People offered MDT back pain rehabilitation and concerned about ability to work due to low back pain. | Semistructured interviews. | Thematic analysis. |
| 8 | Coutu et al,[37] 2010 | Canada | Persistent musculoskeletal pain—back pain (10), upper extremities (4), mixed (2). | 10 men and 6 women aged 25–56 (mean age 40). | Workers referred to work rehab programme. | Semistructured interviews. | Narrative approach—content analysis. |
| 9 | Coutu et al,[60] 2011 | Canada | MSD-related pain for more than 12 weeks accepted and compensated by Quebec Workers' Compensation Board. Back pain (10), upper extremities (4), mixed (2). | 16 workers: 10 men, 6 women, aged 25–56 years (mean age 40). | Referred to evidence-based work rehab programme by a third-party payer. | Semistructured interviews. | Content analysis. |
| 10 | Coutu et al,[61] 2013 | Canada | MSD-related pain for more than 12 weeks accepted and compensated by Quebec Workers' Compensation Board. Back pain (8), upper extremities (2), both (2). | 12 workers (8 men and 4 women) aged 25–56 (mean age 31) and 5 clinicians. | Participants from workers starting a work rehab programme at a hospital research centre. | Multiple case study design: semistructured interviews with workers and rehab clinicians at four points in time. | Thematic analysis and constant comparison method (grounded theory). |

Continued

**Table 1** Continued

| | Author and year of publication | Country | Type of pain | Number, gender and age of participants | Participants | Methods of data collection | Methodological approach—analysis |
|---|---|---|---|---|---|---|---|
| 11 | Crooks,[43] 2007 | Canada | MSD (fibromyalgia, arthritis, RA, OA, lupus). | 18 women aged 26–69 (mean age 44). | Women who developed MSD while involved in the labour market. | Indepth interviews. | Thematic analysis. |
| 12 | Dionne et al,[50] 2013 | Canada | Work-disabling back pain. | Workers with work-disabling back pain. 9=returned to work (7 men, 2 women) aged 30–59. 10=not returned or recently returned (7 men, 2 women) aged 30–60+. | Recruited through newspaper adverts. | Focus groups (2). | Content analysis. |
| 13 | Edén et al,[52] 2007 | Sweden | MSD (type not specified). | 17 individuals (2 men, 15 women) aged 41–62 years. | People going back to work by means of the Swedish 'resting disability pension'. | Interviews. | Inductive analysis relevant to research question. |
| 14* | Fassier et al,[38] 2015 | France | Low back pain. | 3 employers, 1 manager and 1 worker. | Recruited from workplaces with high rates of absence for low back pain—car maker, association providing home services for the dependent and two university hospitals. | Interviews and focus groups. | Qualitative content analysis. |
| 15 | Gard and Sandberg,[40] 1998 | Sweden | Musculoskeletal pain (shoulder, neck and low back pain) for at least 1 year with a period of at least 4 weeks during that time. | 10 patients (9 women, 1 man) aged 30–54 years (mean age 47). | People sick-listed with musculoskeletal pain. | Interviews with a low degree of structure. | Phenomenological structural analysis. |
| 16 | Glavare et al,[54] 2012 | Sweden | Long-term musculoskeletal pain (whiplash, fibromyalgia, nerve injury (neck), arthrosis of the foot). | 11 (8 women and 3 men) aged 22–58 (median age 39). | Participants in a multiprofessional pain rehab programme followed by a coached work-training programme. | Thematised research interviews. | Grounded theory—constant comparative method. |
| 17* | Grataloup et al,[47] 2016 | France | Supervisors of people with musculoskeletal disorders. | Employees' supervisors (61 charge nurses, and head nurses supervising one or more workers with restrictions for heavy lifting or repetitive movements). | Staff from 3 public hospitals. | 12 focus groups (charge nurses and head nurses separate). | Thematic qualitative analysis by constant comparison, each focus group analysed before the next held. |
| 18 | Hansson et al,[30] 2001 | Sweden | Spine-related pain. | 5 people (4 women and 1 man) aged 51–64 (median 55). | People granted disability pension in 1996. | Interviews conducted as conversations—approach based on symbolic interactionism. | Based on grounded theory. |
| 19 | Hansson et al,[34] 2006 | Sweden | Neck or low back pain (spine-related pain). | 33 (20 women and 13 men) aged 32–61 years (median age 48). | Sick-listed participants. | Qualitative interviews. | Qualitative analysis. |
| 20 | Johansson et al,[62] 1997 | Sweden | Undefined musculoskeletal pain disorders. | 20 female patients aged 21–61. | Women sick-listed due to MSD in urban health centre. | Repeated thematic interviews. | Grounded theory. |
| 21 | Juuso et al,[27] 2016 | Sweden | Fibromyalgia. | 15 women aged 38–64 (median 54). | From a rehabilitation centre (4), associations for rheumatism and FM (11). | Indepth qualitative interviews. | Hermeneutic approach. |
| 22 | Kalsi et al,[31] 2016 | UK | Chronic pain (type not specified). | 17 patients (8 men, 9 women) aged 18–65+, but majority (14/17) were 18–34. | Patients attending a 3-week, high-intensity pain management programme. | Semistructured focus group discussion. | Thematic analysis. |

**Table 1** Continued

| | Author and year of publication | Country | Type of pain | Number, gender and age of participants | Participants | Methods of data collection | Methodological approach—analysis |
|---|---|---|---|---|---|---|---|
| 23 | Kvam et al,[63] 2013 | Norway | Prolonged musculoskeletal pain (unspecified pain in the back, neck and shoulders due to fibromyalgia, arthritis and rheumatism). | 4 men and 6 women aged 26–57. | Volunteers from people undergoing vocational rehabilitation. | Semistructured interviews. | Constant comparative analysis. |
| 24 | Kvam and Vik,[51] 2015 | Norway | Prolonged musculoskeletal pain (unspecified pain in the back, neck and shoulders due to fibromyalgia, arthritis and rheumatism). | 6 women, 4 men aged 26–57. | People undergoing vocational rehabilitation. | Indepth interviews. | Discourse analysis. |
| 25 | Liedberg and Henriksson,[36] 2002 | Sweden | Fibromyalgia. | 39 women aged 35–63 (mean 49.5). | Patients from a pain and rehab centre. | Interviews. | Analysed into categories and subcategories. |
| 26 | Magnussen et al,[64] 2007 | Norway | Back pain. | 12 women, 5 men aged 38–56 years (mean age 49). | Part of a larger study evaluating the effect of a vocational rehabilitation-related intervention. | Three focus groups. | Analysis of themes and subthemes. |
| 27 | McCluskey et al,[65] 2011 | UK | Persistent back pain. | 5 dyads (4 male and 1 female claimants), aged 29–54 years (mean age 40.2). | Disability claimants and their significant others. | Semistructured interviews. | Template analysis. |
| 28 | McCluskey et al,[66] 2014 | UK | Persistent low back pain. | 18 (9 benefits claimants: 5 men and 4 women aged 29–63, mean age 48.1) and 9 significant others (6 women and 3 men aged 21–68, mean age 49.7). | Work disability benefits claimants and significant others. | Semistructured interviews. | Template analysis. |
| 29 | Nilsen and Anderssen,[28] 2014 | Norway | Non-malignant chronic pain (neck and back pain, traffic injuries). | 10 men and 10 women aged 26–63 (in the year 2006), mean age 42.7. | From a specialist pain clinic. | Open-ended interviews. | Narrative analysis. Phenomenological meaning condensation framework. |
| 30 | Nordqvist et al,[49] 2003 | Sweden | Back, neck or shoulder diagnoses. | 13 women and 5 men. | People who in 1985 were 25–34 years old and had a new sick leave spell of at least 28 days. | 5 focus groups. | Grounded theory coding and categorising. |
| 31 | Patel et al,[13] 2007 | UK | Chronic musculoskeletal pain. | 38 patients (15 men, 23 women) aged 29–62 years (mean age 49.4). | Unemployed and in receipt of long-term social welfare benefits. | Indepth, semistructured interviews. | Framework approach and thematic analysis. |
| 32 | Rydstad et al,[32] 2010 | Sweden | Whiplash-associated disorders. | 9 people (5 women, 4 men) aged 32–53 years. | Participants of a work-oriented MDT rehab programme. | Thematised interviews. | Constant comparison method—grounded theory. |
| 33 | Saunders et al,[67] 2015 | Canada | MSK injury. Arm (1), knee (1), back injuries (7). | 9 people (5 women, 4 men) aged 34–56 years. | People with long-term work disability and job loss due to an MSK injury from work rehab and chronic pain programmes. | Interviews (27) with 9 people. | Thematic analysis (phenomenological approach guided by life world concept). |
| 34 | Scheermesser et al,[68] 2012 | Switzerland | Low back pain. | 13 (9 men, 4 women) aged 38–60 years (mean age of men 52, mean age of women 48). | Patients with a Southeast European cultural background attending a rehab centre in Switzerland. | Indepth, semistructured interviews (5) and 2 focus groups. | Qualitative content analysis. |

Continued

**Table 1** Continued

| | Author and year of publication | Country | Type of pain | Number, gender and age of participants | Participants | Methods of data collection | Methodological approach—analysis |
|---|---|---|---|---|---|---|---|
| 35 | Shaw and Huang,[33] 2005 | USA | Occupational low back pain. | Focus group: 28 people (15 men, 13 women) aged 31–65 (mean age 46). Interviewees: 23 people (11 men, 12 women) aged 25–64 (mean age 42.6). | Focus group participants: people recently (<6 months) returned to work after injury responding to newspaper advert. Interview participants: patients referred by physios from collaborating OH network. | Focus group and interviews. | Content analysis. |
| 36 | Sjöström et al,[29] 2011 | Sweden | MSK disorders—mainly back and neck pain. | 10 people (7 women, 3 men) aged 29–61 (mean age 48). | Attended a rehab programme and still on full-time sick leave 2 years after completion. | Semistructured interviews. | Qualitative content analysis. |
| 37 | Soeker et al,[44] 2008 | South Africa | Back injury. | 26 people (18 men, 8 women) aged 18–60. | Selected by random sampling from a hospital rehab department. | Focus groups. | Qualitative analysis. |
| 38* | Soklaridis et al,[41] 2010 | Canada | Low back pain—work-related injury. | 59 stakeholders including 6 injured workers and 5 small and 9 large employers. | Various contacts of the research team. | 9 focus groups. | Grounded theory approach. |
| 39 | Svensson et al,[39] 2010 | Sweden | Back neck or shoulder diagnosis. | 13 women and 5 men. | People aged 25–34 years old in 1985 and had a new sick leave spell of at least 28 days. | 5 focus groups. | Descriptive and explorative method of analysis. |
| 40* | Williams-Whitt et al,[46] 2016 | Canada | Low back pain. | 23 supervisors. | Supervisors of back-injured workers from 11 Canadian organisations. | Semistructured, indepth interviews. | Constructivist grounded theory principles. |
| 41* | Wynne-Jones et al,[35] 2011 | Wales | Musculoskeletal pain. | 18 employees with MSK pain (8 men, 10 women), mean age 49.7. 20 managers (10 men, 10 women), mean age 44.8. | Two large public sector organisations. | Semistructured interviews. | Thematic analysis. |

*Employer studies.
FM, fibromyalgia; MDT, multi-disciplinary; MSD, musculoskeletal disorders; MSK, musculoskeletal; OA, osteoarthritis; OH, occupational health; RA, rheumatoid arthritis.

**Table 2** Conceptual categories, description of category, supporting studies and CERQual assessment

| | Conceptual category— summary of review finding | Quotations from primary studies to illustrate conceptual category | Supporting studies | Adequacy (number of concepts— see list of concepts in online supplementary file 3) | Coherence (number of supporting studies) | Methodological limitations (see CASP scores in online supplementary file 1) | Relevance | Overall CERQual assessment of confidence in the evidence | Explanation of CERQual assessment |
|---|---|---|---|---|---|---|---|---|---|
| 1 | *Managing pain*—the impact of pain on return to work and how it can be managed. | Chronic pain itself was the underlying barrier from which most other barriers to work stem. Overall, very few patients reported any attempts to plan for the future, primarily due to the unpredictable nature of the pain condition and physical mobility problems associated: *"I have no objections at all to go back to work. But, I thought about this. I don't know what I could do. I can't sit for very long. I can't stand for very long. Erm, in discomfort 99% of the time."* (Male, 56)[13] | 12 13 27-37 40 48 50 54 58 65 68 | Richly described (49). | Fit between underlying data and review finding is very clear (20). | All CASP scores over 20. | Sweden 9 UK 5 Canada 2 Norway 1 Switzerland 1 USA 1 Australia 1 3 studies partially relevant as only included women not men. | High confidence. | Graded as high in relation to adequacy, coherence, methodological limitations and relevance. Employee and employer studies support this finding. |
| 2 | *Managing work relationships*—impact of chronic pain on relationships with employers, line managers and colleagues and how this is managed. | The existence of interpersonal conflict with colleagues or managers was mentioned as a barrier, as was mutual mistrust. For managers, overwork, role conflict between production targets and occupational health, and a lack of hierarchic support were possible barriers. With colleagues, overwork and scepticism about medical problems could induce hostility and rejection. For workers with LBP, the feeling of being judged and having to justify absence, pain and limitations were perceived as a barrier.[38] | 12 13 27 31 32 35 36 38-50-52 54 62 64 67 | Richly described (51). | Fit between underlying data and review finding is very clear (25). | All CASP scores over 20. | Sweden 10 Denmark 1 UK 5 Canada 5 France 2 Norway 1 South Africa 1 | High confidence. | Graded as high in relation to adequacy, coherence, methodological limitations and relevance. Employee and employer studies support this finding. |
| 3 | *Making workplace adjustments*—the scope and process for making changes to the job, work conditions or environment. | Our data suggest that the ability of participants to remain in employment was in part influenced by the nature of their work (whether or not adaptations could be made to enable employees to continue in part despite their symptoms) and in part due to patients' confidence and ability to negotiate adaptations with their employers (significant others often described themselves as being an important source of support for the patient in this context).[48] Accommodation management is perceived as a considerable addition to supervisors' regular duties, for which they feel ill-prepared, even where guidance is provided by others with the requisite expertise.[46] | 12 13 28 29 34 36 38 41 43-53 59 | Richly described (55). | Fit between underlying data and review finding is very clear (20). | All CASP scores over 20. | Denmark 2 Sweden 6 UK 3 Canada 4 Norway 2 France 2 South Africa 1 | High confidence. | Graded as high in relation to adequacy, coherence, methodological limitations and relevance. Employee and employer studies support this finding. |
| 4 | *Autonomy*—the individual's ability to have control or agency in relation to their pain and their work situation. | Several respondents emphasised the importance of their possibilities to control what to do and to do it and considered flexible working hours as a prerequisite for their return to work.[52] | 12 13 30 31 40 45 48 50 52-54 59 | Well described (12). | Fit between underlying data and review finding is clear (12). | All CASP scores over 20. | Sweden 5 UK 4 Denmark 2 Canada 1 | Moderate confidence. | Graded as moderate as well described and relevant across four cultures. |
| 5 | *Self-belief/self-efficacy*—the individual's outlook about their ability to handle work and manage their pain. | Self-efficacy statements pertaining to more complex work-related functions were subdivided into one of three categories based on the thematic content analysis: the ability to meet job demands, the ability to obtain help from others and the ability to cope with pain.[33] | 12 31-33 39 40 48 52 53 63 64 | Richly described (18). | Fit between underlying data and review finding is very clear (11). | All CASP scores over 20. | Sweden 5 Denmark 1 Norway 2 UK 2 USA 1 | High confidence. | Graded as high in relation to adequacy, coherence, methodological limitations and relevance. |
| 6 | *Being believed*—people struggling with not being believed, trusted or perceived as legitimate. | Employees typically discussed issues around being believed and trusted when they were ill. Managers, on the other hand, were more likely to talk about employees taking absence that was not legitimate. For example: *"People just don't turn up, you know. They phone in sick or er… The attitude is they, you know, 'why should I bother?' sort of thing. You get a lot of that."* (Female manager)[35] | 32 33 35 38 44 53 54 59 65 | Well described (13). | Fit between underlying data and review finding is clear (8). | All CASP scores over 20. | Denmark 2 France 1 Sweden 2 UK 2 South Africa 1 | Moderate confidence. | Graded as moderate as well described and relevant across five cultures. Employee and employer studies support this finding. |
| 7 | *Impact of and on the family*—the effects of chronic pain on family members and vice versa. | Lots of patients with LBP are looked after by their family members who relieve them of physical activities. Many patients receive more attention and are encouraged to take rest. The positive feeling of being supported is counteracted by the negative feeling of uselessness, associated with being off work.[68] | 32 36 48 51 58 65 66 68 | Well described (13). | Fit between underlying data and review finding is clear (8). | All CASP scores over 20. | Australia 1 UK 3 Sweden 2 Switzerland 1 Norway 1 | Moderate confidence. | Graded as moderate as well described and relevant across five cultures. |
| 8 | *Not being understood*—this is in the context of relationships with health professionals. | Participants felt that physicians did not understand their clients' work environment, such as what functional demands were necessary for them to complete their tasks, as well as the psychosocial stressors that could cause their back pathology to become chronic.[44] | 28 41 44 51 61 68 | Adequately described (9). | Some inconsistency in fit (6). | All CASP scores over 20. | Canada 2 Norway 2 Switzerland 1 South Africa 1 | Low confidence. | Graded as low as some concerns about coherence and relevant across four cultures. Employee and employer studies support this finding. |
| 9 | *Finance and benefits*—financial difficulties and the economic insecurity of moving from welfare benefits back into work. | Some patients who were unemployed on grounds of ill health had serious concerns about their financial future. They complained of sleep disorders and mental problems: *"Without medication I can't sleep. I don't know what is going to happen."*[68] *"It must be possible to find transition solutions when trying to get back to work, solutions that make us feel economically secure. Otherwise, who would dare to try?"*[64] | 13 30 33 36 49 62 64 68 | Well described (10). | Fit between underlying data and review finding is clear (8). | All CASP scores over 20. | Canada 1 UK 1 USA 1 Switzerland 1 Sweden 3 Norway 1 | Moderate confidence. | Graded as moderate as well described and relevant across six cultures. |

Continued

**Table 2** Continued

| | Conceptual category—summary of review finding | Quotations from primary studies to illustrate conceptual category | Supporting studies | Adequacy (number of concepts—see list of concepts in online supplementary file 3) | Coherence (number of supporting studies) | Methodological limitations (see CASP scores in online supplementary file 1) | Relevance | Overall CERQual assessment of confidence in the evidence | Explanation of CERQual assessment |
|---|---|---|---|---|---|---|---|---|---|
| 10 | Health and illness and pain representations impact on return to work—the way people think about their pain and the mental representations they form in relation to beliefs about its cause and their perception of its impact on their lives. | Beliefs about causality. All claimants reported work as the initial cause of their back pain condition, and most also perceived previous work/certain types of work (manual/heavy/repetitive) as a 'trigger' for subsequent episodes and therefore not conducive to return to work.[65] Both patients and significant others in the non-working sample were resigned to the permanent effects of the patient's back problem on their employment status and were thus more likely to consider the patient as 'disabled', a role which might become self-fulfilling.[48] | 27 31 32 37 41 48 53 54 58–60 65 68 | Richly described (22). | Fit between underlying data and review finding is very clear (13). | All CASP scores over 20. | Denmark 2 Australia 1 UK 3 Canada 3 Sweden 3 Switzerland 1 | High confidence. | Graded as high in relation to adequacy, coherence, methodological limitations and relevance. Employee and employer studies support this finding. |
| 11 | Meaning of work—the meaning of work for an individual linked with motivation. | Work was viewed by most patients as a source of financial security and a means of independence. However, the financial aspect of work did not seem to motivate all patients. A separate subgroup placed greater focus on health status and pain reduction strategies. This was a more prevalent attitude among those who had had longer durations of sick leave. For example, a patient who had been unemployed for more than 1 year stated: "Money wouldn't be a motivator for me, I'd have to be well enough to return to work – that would be the motivator."[31] Participants who were placed in jobs that had no meaning to them caused them to become frustrated. When these meaningless tasks were coupled with relapses of back pain, there would be a downward spiral into depression and demotivation among the participants.[44] | 27 28 31 34–36 39–41 44 46 52 54 62 63 67 | Richly described (26). | Fit between underlying data and review finding is very clear (14). | All CASP scores over 20. | Sweden 7 UK 2 Norway 2 Canada 2 South Africa 1 | High confidence. | Graded as high in relation to relevance, coherence, adequacy and methodological limitations. Employee and employer studies support this finding. |
| 12 | Mismatch between employee and employer expectations of return to work. | 'Meeting job demands typically referred to producing a certain quantity of work (eg, 'I need to be at full capacity'), quality of work (eg, 'I may not do a good job'), speed of work (eg, 'I won't be able to keep up') or fulfilling a particular role at work (eg, 'I need to be able to respond to an emergency').'[33] | 13 29 31 33 40 46 49 52 63 64 | Well described (11). | Fit between underlying data and review finding is clear (10). | All CASP scores over 20. | Sweden 4 UK 2 Norway 2 USA 1 Canada 1 | Moderate confidence. | Graded as moderate as well described and relevant across five cultures. Employee and employer studies support this finding. |
| 13 | Social isolation as a consequence of chronic pain—leading to a lack of support to return to work. | 'Paid work – the pain sufferer's struggle for social capital. The informants were concerned about how the unpredictability of the pain broke into their daily lives and social contact with others, and challenged their normal way of dealing with everyday problems'.[28] | 28 32 36 58 59 | Adequately described (6). | Fit between underlying data and review finding is clear (5). | All CASP scores over 20. | Australia 1 Denmark 1 Sweden 2 Norway 1 | Low confidence. | Graded as low as adequately described and relevant across four cultures. |
| 14 | Influence of return to work support and rehabilitation. | Support. The informants felt that the rehabilitation programme was the right place to come to when living with long-term pain. The team was described as empathetic and knowledgeable: when the informants told about their difficulties they felt understood for the first time. The informants also got support from each other, and a good feeling of fellowship developed; on the other hand, some also described how they were negatively affected by other participants who were depressed.[32] | 31–33 36 37 42 50 52–54 59 61 68 | Well described (19). | Some inconsistency in fit (13). | All CASP scores over 20. | Denmark 3 Canada 3 Sweden 4 Switzerland 1 USA 1 UK 1 | Moderate confidence. | Graded as moderate as well described and relevant across six cultures. |
| 15 | System factors (healthcare, social security and workplace systems)—how policies and procedures in these three systems impact on people with chronic pain. | The way 'systems' (dys)function delays return to work. When a worker becomes injured, they enter into complex relationships with the compensation system, unions, workplace and healthcare system. How these systems interact with one other and with the injured worker can affect the RTW process.[41] Within the healthcare system, participants talked of their frustrations with long waiting times, such as waiting for specialist appointments, diagnostic tests, treatments and entry into programmes.[67] 'De-motivating economic arrangements for pensioners were also mentioned as a barrier. Some pointed to the fact that income under re-education was so low that the effort was not worthwhile. In addition, several had experienced that taking a small part time job when receiving pension would reduce the benefit so that nothing was gained economically. Finally, trying out for new jobs arranged by the job centre put them in an economically uncertain position and made them afraid of losing their disability benefit all together. More appropriate transitional arrangements were asked for'.[64] | 13 32 33 38 41 43–45 50 54 64 67 | Richly described (25). | Fit between underlying data and review finding is very clear (12). | All CASP scores over 20. | UK 3 Canada 4 Sweden 2 Norway 1 South Africa 1 France 1 | High confidence. | Graded as high in relation to relevance, coherence, adequacy and methodological limitations. Employee and employer studies support this finding. |
| 16 | Societal expectations—expectations of family, friends and wider society that everyone should work, resulting in judgement and discrimination against those who do not work. | Experiences of societal expectations of participation in work. In the societal discourse of work participation, inclusion in society was connected to employment.[51] Some of the participants viewed their family and society as being judgemental, unsupportive and discriminatory, whereas others felt that they could not have rehabilitated themselves without the support of society.[44] | 44 51 52 63 | Adequately described (3). | Some inconsistency in fit (4). | All CASP scores over 20. | Sweden 1 Norway 2 South Africa 1 | Low. | Graded as low as adequately described and relevant across three cultures. |

NB: No more than half female-only studies supported any of the review findings.
CASP, Critical Appraisal Skills Programme; CERQual, Confidence in the level of Evidence from Reviews of Qualitative research; LBP, low back pain; RTW, return to work.

and the way they are influenced by other factors appear to be central to negotiating a successful return to work. The scope for managing pain and making adjustments in the workplace can be influenced by the quality of the relationship an individual has with their employer and/or line manager and what is feasible within a particular work setting. The remaining 13 conceptual categories are described in more detail in online supplementary file 3. The concepts within each conceptual category are presented in online supplementary file 4.

### Managing pain

Pain was seen as a major obstacle to return to work.[27–29] A plethora of strategies to manage it were described,[28–33] including use of sick leave.[12 34]

> They used the strategies doing a little at a time, taking continuous breaks, working slower and being aware of body posture and workloads. These strategies improved their endurance and prevented further pain.[32]

> However, the strain of living with chronic pain meant fatigue also became a problem and low-energy levels prevented work return.[29]

> Pain developed and became continuous, was easily provoked by work tasks and relatively resistant to pain-controlling strategies. Life became strenuous and energy was reduced.[30]

The impact of pain on performance[35] and ability to attend and travel to work,[36] along with the fear of pain exacerbation,[31 37] were also problematic.

### Managing work relationships

Interpersonal conflict and mutual mistrust can arise between people with pain and their employers and colleagues,[38 39] and if relationships with supervisors are perceived as poor then this is demotivating in relation to work return.[40] Employers with few employees expressed reservations about how far to push an employee for fear of upsetting them and causing them to be off sick for longer than necessary.[41] Managers in a public sector study appeared to be walking a fine line between supporting employees, making sure colleagues were not adversely affected and that services were delivered.[35] Asking for help was perceived as frustrating by people in pain, and incurred feelings of inadequacy and negativity.[42] Some struggled in their interaction with employers and tended to be passive, not believing their views were listened to, or valued, which led to difficulties in sustaining work return.[12] Unsympathetic employer attitude and a lack of understanding of the person's experience of pain were seen as major obstacles to work return,[13 31 32 43 44] but those employers with personal experience of pain were perceived as more sympathetic and empathic.[45 46]

> One of the important employment related obstacles is the perception that employers have limited understanding about pain due to ignorance and a lack of awareness. However, patients do acknowledge that chronic pain is difficult to understand without personal experience.[13]

Team management responsibilities of regulating tension between colleagues were perceived as challenging when work restrictions for those with pain caused unequal work distribution, leading to a sense of injustice.[47]

> However, if duties were reduced indefinitely, with no extra cover, workers might feel that they were burdening their colleagues. There were doubts as to how long their colleagues support might continue.[45]

### Making workplace adjustments

An economic climate of austerity was perceived as an obstacle to work due to reduced job availability and a competitive job market.[13] Reorganisations and rationalisation in the workplace meant jobs had changed and become more demanding and potentially difficult to adapt for people with a pain condition.[36] In this situation, age was also seen as influential, with some feeling they were too old to retrain for a different kind of job.[13 44]

The type of job influenced work return decisions, with physical work being perceived as more challenging with pain[44 47] and more highly skilled work providing greater scope for flexibility and adaptation.[48]

> Modifying work hours and days is a potential accommodation for women who develop musculoskeletal diseases, but it is only appropriate in certain work environments where such flexibility is allowed.[43]

People with chronic pain often felt they were not consulted or involved in the decision making about workplace redeployment or adjustment, and when desired modifications were not possible they could not return to work.[43–45 48] Managers' attitudes and efforts,[46] combined with effective routine methods of regular communication of changes made to colleagues,[49] were seen as ways of improving the success of workplace adjustments. Managers did not always have the resources or know what options would be available for making these adjustments and saw the planning of these accommodations as an additional demand on their time.[46] Managers also felt that information about work restrictions from occupational health was not always realistic in the work setting and therefore difficult to implement.[47]

> Many charge and head nurses complained that occupational physicians formulated unrealistic restrictions that were impossible to respect due to work organization.[47]

A number of workplace adjustments were felt to be helpful, including flexible hours or a reduction in hours, but were not always forthcoming.[36 43 48] The possibility of a gradual return to work,[50] working from home or participating in a job sharing programme[43] was also seen as helpful by people in chronic pain. Changes to the job itself, including physical adjustments and a reduction in

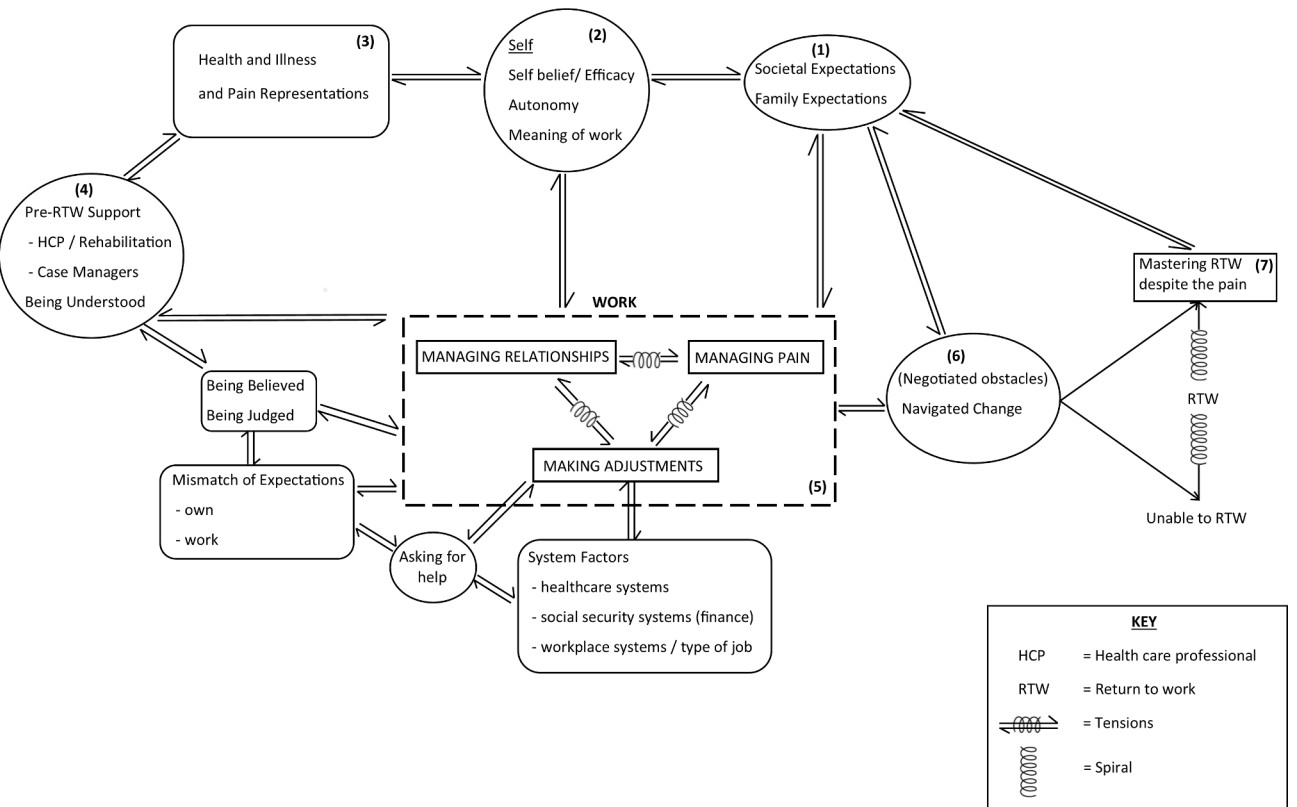

**Figure 2** Conceptual model: the work of return to work. This conceptual model of return to work is explained in the text, going anticlockwise from (1) to (7).

job demands, were not always feasible, for example in a nursing,[43] nursing assistant role[36] or a preschool teaching role.[51]

### Line of argument
A line of argument was constructed by examining how the conceptual categories relate to each other. A flow diagram/conceptual model was then developed (figure 2).

This conceptual model of return to work is now explained, going anticlockwise from (1) to (7).

1. The underpinning foundation lies in the cultural expectation within society that people should work and contribute to the economy. Societal expectations are manifested within institutions, families and the media.
2. Societal and family expectations influence the individual's sense of self and what work means to each person. Meaning can relate to financial remuneration, rewards or survival, and meeting of social, cognitive and achievement needs or purpose in life. The individual's level of self-belief and autonomy will both play a part in how much agency and control can be exerted over pain and the work situation.
3. The way someone thinks about their pain and the mental representation they create will also influence their behaviour and the possibility of returning to work. People's perceptions of whether pain is a long-term disability could influence whether they feel able to work and thus their return to work decisions, where-

as someone who has accepted the pain as part of their life and adapts may be more likely to consider return to work.
4. Some studies in the review evaluated prereturn to work support or rehabilitation programmes, and not being understood by health professionals was cited as an obstacle. In the same way, not being believed or being judged by people in the workplace was also perceived to make return to work challenging.
5. The three key tenets of return to work are managing pain, managing work relationships and making workplace adjustments. Tension exists between these three facets and they can be influenced by a mismatch between the individual and the employer expectations, difficulties asking for help and system factors in the workplace, and health and social security systems.
6. Individuals must negotiate a wide range of obstacles and navigate change.
7. This could result in a downward spiral (and not working) at one end of a continuum through to an upward spiral of mastering return to work despite pain.

### DISCUSSION
In this meta-ethnography we identified obstacles to return to work for people with chronic pain centred around three key conceptual categories: managing pain, managing work relationships in the workplace and making workplace adjustments. The dynamic relationship

between these three closely linked categories appears to be highly influential in navigating change and overcoming obstacles individuals with chronic pain face. The ability to manage pain and negotiate workplace adjustments can be affected by the strength of relationships with employer and colleagues and what is practicable in the work environment.

The concepts of health and pain representations and the role of significant others and their thinking about pain and return to work do not appear to be highlighted by previous reviews. Another neglected area is the influence of prereturn to work support or rehabilitation. The employer perspective is missing in earlier reviews which have focused on the experience of people with chronic pain. Only five of the included studies were conducted with employers, so there is still limited research with this group despite people with pain emphasising the importance of employer attitudes and knowledge in the return to work process.[49]

Some studies that were included in the review appear to suggest that those people with chronic pain who manage to stay in work have different characteristics from those who are unable to do so. This is seen in part to be connected with their cognitive appraisal of their pain and whether they are able to adapt.[31 37] It has been proposed that in those who do not return to work, their pain representation of 'abnormal pain' becomes crystallised with their goal of pain elimination firmly intact, whereas those who returned to work began to perceive pain as 'the new normal' and something they learn to live with.[37] Edén et al[52] described three different adaptation patterns: the go-getter, realist and indifferent. They proposed the pessimistic and passive outlook of the latter type meant work return was less likely. Passivity in the interaction with stakeholders like the employer was found to be linked with reduced drive to return to work.[12] Angel et al[53] and Dionne et al[50] also found passivity in relation to pain was not helpful when addressing workplace obstacles.

The provision of professional individualised support and coaching in the workplace was seen to be valuable in the work return process,[54] and this concept supports the idea of developing work-based interventions to help people with chronic pain return to work.

When comparing findings with previous reviews that have highlighted obstacles to return to work, similarities include fears of not being able to fulfil employer expectations, not being believed by colleagues and financial concerns.[4] Worries for the future, including financial and job security, were also uncovered by MacNeela et al.[5] Strain on the family relationships, including those with partners and children,[4] and gender differences regarding role as carer or breadwinner were revealed.[5] Unsatisfying relationships with health professionals where people felt they were not being listened to and frustrations with limitations of medical treatment were other common features.[5] Social withdrawal as a result of pain was highlighted in both of these reviews.[4 5] A struggle for legitimacy with colleagues and stigma in the workplace was highlighted

by Toye et al[7] and Froud et al.[4] This review also drew attention to the system not supporting return to work due to a lack of dialogue between employers, occupational health and the health system to facilitate a gradual return with appropriate adjustments.

The collaborative team approach to conceptual analysis increased the rigour of the review.[14] Independently drawing flow diagrams to illustrate the conceptual model and then coming together to amalgamate these through discussion and debate, combined with checking all concepts had been included, ensured this process was thorough.

The CERQual assessments indicated there was a high level of confidence in the findings for managing pain, managing work relationships, managing the workplace, self-belief, health and illness representations, the meaning of work and system factors. Although we have used CERQual, we found we agreed with many comments on its use by Toye et al,[55] namely that for relevance, studies rated as partially or indirectly relevant could also contain helpful concepts. They suggest 'gravitational pull' of an idea may be important. They argue providing clear information about concepts is critical, and we have provided this in online supplementary files 3 and 4. They also note for adequacy. 'The power of concepts to make us think, however, is not based on quantity of data included'. We agree when looking at coherence that inconsistent findings do not necessarily call the findings into question. It may be one study has developed an insight not considered in other studies. No tool can guarantee confidence in findings, and authors still need to carefully consider rigour issues.

## Implications

This review identifies obstacles faced by people with chronic pain in returning to work after a period of sick leave or unemployment and can be used to inform the development of a return to work intervention. The focus of such intervention should be working collaboratively with the person who has chronic pain and the employer to explore ways of addressing managing pain, managing work relationships and making workplace adjustments. The way in which the different factors work together either to enhance or inhibit return to work is highly individual, and clinicians will need to assess what is most important for the person and employer with whom they are working. This intervention could be located in community/primary healthcare and delivered by case managers, for example, occupational therapists or occupational health nurses working alongside general practitioners. Alternatively it could be delivered by employment specialists working in employment services and trained in pain management strategies. This type of intervention would provide support tailored to the specific needs of people with chronic pain. Discussion may be needed between the employer, the employee and the case manager to enable exploration of the ways in which obstacles to return to work might be overcome. This collaborative approach

has the potential to improve healthcare services and change workplace culture and is the kind of innovation envisioned by the UK government in their 10-year plan for people with long-term health conditions to realise their working potential.[56]

## Limitations

It is apparent that more research is required from the employer's perspective. The five studies included in the review were from the perspective of employers working in car making, university hospitals, home care provision for disabled people in France,[38] public hospitals in France,[47] and National Health Service Trust and local authority in Wales.[35] The Canadian study that included small and large employers did not specify the nature of the industry in which they were engaged.[41]

It is likely that the reviewers' backgrounds and experiences had an impact on synthesis findings. The authors came from healthcare professional and non-healthcare professional backgrounds, and these backgrounds and experiences of chronic pain provided certain lenses, which we would expect to influence our understanding.

At the time we did this work, the eMERGe Reporting Guidance for meta-ethnography[57] had not been published. They were published close to the end of the peer review process for this paper.

## CONCLUSIONS

The navigation of obstacles to return to work for people with chronic pain and their employers entails balancing the needs of the person with chronic pain, colleagues and the employing organisation. The influence of health and pain representations the person formulates has not been emphasised in previous reviews. Managing pain, managing relationships in the workplace and making adjustments are central to achieving a successful return to work, and these can be hard work for the person with chronic pain.

**Acknowledgements** We would like to thank Samantha Johnson, Academic Support Librarian (Medicine, Life Sciences and Psychology) at the University of Warwick, for her help with initial literature search and helpful guidance during the updated search process. We would like to thank Debs Smith, patient and public involvement representative, who participated in developing the research protocol as a coapplicant.

**Contributors** KS, RF and MU made a substantial contribution to the design of the study. MG and KS were responsible for acquisition, and MG, KS and JO-B-E were responsible for analysis and interpretation of the data. MG drafted the first, subsequent and final versions, and KS, JO-B-E, RF and MU revised all versions for important intellectual content and approved the final version. All authors agree to be accountable for the accuracy and integrity of the work.

**Funding** This research is part of the RISE (Return to work with Individualised Supported Employment) feasibility study supported by the Versus Arthritis charity (project number 9401), and the funder has had no involvement in this paper.

**Competing interests** RF is chief investigator on the Versus Arthritis grant from which this project was funded. He has published multiple papers on chronic pain, some of which are referenced in this paper. RF and MU are part of an academic partnership with Serco related to return-to-work initiatives. RF and MU are directors and shareholders of Clinvivo, a university spin-out company that provides data collection services for health services research. MU was Chair of the NICE

Accreditation Advisory Committee until March 2017 for which he received a fee. He is chief investigator or coinvestigator on multiple previous and current research grants from the UK National Institute for Health Research and Versus Arthritis, and is a coinvestigator on grants funded by the Australian NHMRC. He is an NIHR Senior Investigator. He has received travel expenses for speaking at conferences from the professional organisations hosting the conferences. He is a coinvestigator on a study receiving support in kind from OrthoSpace. He has accepted an honorarium from CARTA. He is an editor of the NIHR journal series, and a member of the NIHR Journal Editors Group, for which he receives a fee. KS is an investigator in multiple previous and current research grants from the UK National Institute for Health Research and Versus Arthritis. She has received travel and accommodation expenses for speaking at conferences from the professional organisations hosting the conferences. She has published multiple papers on pain, some of which are referenced in this paper.

**Patient consent for publication** Not required.

**Provenance and peer review** Not commissioned; externally peer reviewed.

**Data sharing statement** This is a qualitative systematic review, so there are no primary research data. We have tried to include all relevant data for the qualitative systematic review in the supplementary files. Any other reasonable requests will be considered on a case-by-case basis by MG (lead author; email: M.Grant.2@warwick.ac.uk).

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
