## [Reviewer comments · BMJ Open]

ARTICLE DETAILS

TITLE (PROVISIONAL)	The work of return to work. Challenges of returning to work when you have chronic pain: a meta-ethnography
AUTHORS	Grant, Mary; O-Beirne-Elliman, Joanne; Froud, Robert; Underwood, Martin; Seers, Kate

VERSION 1 - REVIEW

REVIEWER	Gwenllian Wynne-Jones Arthritis Research UK Primary Care Centre Keele University UK
REVIEW RETURNED	23-Oct-2018

GENERAL COMMENTS	This is a very interesting and timely paper clearly highlighting the difficulties in returning to work with chronic pain. The introduction spells out the rationale for the study leading to clearly defined aims. The concept and use of the meta-ethnography methodology is well described as are the methods used to conduct the systematic review. The findings are well presented with conceptual categories derived from the papers identified through the systematic review, with a conceptual model developed to illustrate the relationship between these categories. There is a good discussion, setting the findings in the context of other literature with acknowledgement of the limitations of the current work and consideration of the implications. The authors state that the search terms used "included...", it would be helpful if the search strategy could be included either in a supplementary file or a note added to the text that the strategy is available on request. The analysis of papers was principally carried out by one author, with two others checking the first ten papers (alphabetically). It is unclear whether this was a percentage of the papers identified or the rationale for selecting just 10 papers, could the authors justify this approach please? It would also be helpful to have a sense of how much agreement there was in identifying the third order concepts, it is reported that this took place over three meetings but was there any overlap in the concepts initially identified? The CASP tool was used to appraise the included papers, it is noted that all included studies passed the first two screening questions, were papers excluded if they did not pass these screening questions?
---

	There is a short section on the implications of the paper, I feel that this section would benefit from some further consideration. The authors suggest developing a return to work intervention to explore ways of addressing and managing chronic pain, managing work relationships and making workplace adjustments, but who would deliver such an intervention? Should it be located in healthcare, with the employer, another agency, or a combination? What are the likely implications of such an intervention on healthcare services and employers (improving health/changing workplace cultures)?
--	--

REVIEWER	Hazel Keedle Western Sydney University Australia
REVIEW RETURNED	27-Oct-2018

GENERAL COMMENTS	This is a very interesting manuscript using the methodology of meta-ethnography. I have only found a few sections that I have some comments or recommendations for: Line 136 - not sure if identifying the 11th version is necessary or relevant. You have identified that there were several revisions and I believe that is adequate. Line 136-139 - the sentence describing the methodology used by Toye et al doesn't read well, even just removing the 'and this' would help. Line 144 - how was the focus endorsed by the representative, when reading that it makes me want to have more information. Table 2 - I really liked this table as it clearly shows the relevance of the conceptual categories within the supporting studies. Although you have put quotations from the primary studies in this table I find that they are notably missing from the results section. I think incorporating different examples of quotes in the key conceptual categories descriptions would add more depth to the concepts. Starting line 168 - There needs to be more information given here on the relevance of the three key conceptual categories as this is missing. You have a great explanation of them in lines 260-266 at the beginning of the discussion but I think a bit more is needed earlier. This paragraph from the discussion would even fit well in this first section of the overarching conceptual categories to give the reader a good explanation of the relevance and importance of the three key conceptual categories. Figure 2 - conceptual model - This is a good model but it does seem unnatural going anti-clockwise and then this needs to be identified as you have done as anti-clockwise. Line 303 - instead of saying 'carefully reading and re-reading abstracts and then full texts' it might have more consistency if you said for example 'by applying the seven steps of the meta-ethnography process'. Otherwise there is a potential confusion over what methodology was used. Line 340-341- the sentence on the influence of health and pain is overly wordy and confusing, but this isn't my subject area of expertise so it might be correctly pitched.
---

	Thank you for the opportunity for reviewing this interesting meta-ethnography.
--	--

REVIEWER	Professor Nicky Britten College of Medicine and Health, University of Exeter, UK I am a co-author of the recently published paper by Emma France and colleagues (EF France, M Cunningham, N Ring, I Uny, EAS Duncan, RG Jepson, M Maxwell, RJ Roberts, RL Turley, A Booth, N Britten, K Flemming, I Gallagher, R Garside, K Hannes, S Lewin, GW Noblit, C Pope, J Thomas, M Vanstone, GMA Higginbottom, J Noyes. Improving reporting of Meta-Ethnography: The eMERGe Reporting Guidance, Journal of Advanced Nursing, DOI: 10.1111/jan.13809, 15 January 2019) which was simultaneously published in an open access format in several journals.
REVIEW RETURNED	30-Jan-2019

GENERAL COMMENTS	The work of return to work The question of return to work by those people suffering chronic pain is clearly of great significance to the individuals involved, their employers, and the economy as a whole, not to mention doctors who write sick notes. This paper tackles an important question by aiming to synthesise qualitative research, using the method of meta-ethnography, from the point of view of both employees and employers. I have no expertise in either chronic pain or return to work (or writing sick notes), but provide a review from a methodological perspective. To declare my own interest, I am a co-author of the recently published paper by Emma France and colleagues (EF France, M Cunningham, N Ring, I Uny, EAS Duncan, RG Jepson, M Maxwell, RJ Roberts, RL Turley, A Booth, N Britten, K Flemming, I Gallagher, R Garside, K Hannes, S Lewin, GW Noblit, C Pope, J Thomas, M Vanstone, GMA Higginbottom, J Noyes. Improving reporting of Meta-Ethnography: The eMERGe Reporting Guidance, Journal of Advanced Nursing, DOI: 10.1111/jan.13809, 15 January 2019) which was simultaneously published in an open access format in several journals. I will draw on the eMERGe Reporting Guidance in writing this review. The authors of the current paper (to be referred to as 'synthesis authors' to avoid confusion later) may be helped by the explanations of the 19 reporting criteria given in the Guidance. The paper under review also used the GRADE-CERQual guidance with which I am less familiar to assess the strength of evidence in different conceptual categories. The rationale and context for using the method of meta-ethnography is clearly described as are its aims and focus. In particular the synthesis authors cite knowledge gaps in relation to return to work by people with chronic non-malignant pain, and the perspective of employers. However readers of BMJ Open may also be interested in the perspectives of doctors: perhaps the synthesis authors could at least indicate if there is any qualitative literature on this topic. In relation to the aim of the synthesis, the synthesis authors have gone beyond 'understanding obstacles' (abstract, page 2) and have provided a conceptual model of return to work. Thus it might be appropriate to revise the aim of the synthesis to reflect its more ambitious outcome (eg 'to provide a conceptual model'). The synthesis authors explain why they considered meta-ethnography to be the most appropriate qualitative synthesis methodology, saying that they wished to
--

'identify any other overarching concepts that would explain the data' (page 5).

The synthesis authors list the databases searched and the search terms used but they do not describe the rationale for their search strategy (for example, why these databases were considered the most appropriate ones to use). They seem not to have used a search strategy specifically developed for qualitative studies (as discussed by Booth 2016 for example), which might perhaps have identified other qualitative papers. They do not say who conducted the screening and selection of papers although possibly these were the same people who did the searching. Some of the exclusion criteria are given in the results section.

The results of the searches and screening are clearly and fully set out in figure 1. The synthesis authors explain who read the 41 included papers, how the data were extracted and the conceptual categories identified. The synthesis authors state that 'the concepts are ideas drawn from the findings of the original papers' (page 6) though they do not distinguish between participant quotes and authors' findings (often referred to as first and second order concepts, see for example Britten et al, 2002). The first column in table 2 seems to contain the synthesis authors' summaries (often referred to as third order concepts) and the second column contains participants' quotes (first order concepts). Thus it is not clear if the synthesis authors have worked with second order concepts. The characteristics of the included studies, and the data extracted from each study, are clearly set out in Table 1. However there is no discussion of which aspects of the included studies were compared, nor how the studies were compared. Noblit and Hare (1988), whose approach the synthesis authors cite, refer to reciprocal and refutational syntheses, but it is not clear that the synthesis authors considered the methods of translation in these terms. Thus, point 13 of the eMERGe Reporting Guidance referred to above asks about the steps taken to preserve the context and meaning of relationships between concepts within and across studies; how the reciprocal and/or refutational translations were conducted; and how potential alternative interpretations and explanations were considered. It might have been helpful, for example, to group the 5 papers covering employer perspectives separately; or to consider if the differences between employees and employers merely reflect different perspectives or if they constitute refutational translations. The three key conceptual categories are described in the text while the remaining 13 categories are described in the supplementary material, no doubt for reasons of space, as well as briefly in table 2.

The synthesis authors describe the ways in which they developed overarching concepts and in particular how three of them independently developed their own conceptual models before comparing them and eventually agreeing a final (11th) version. It is not stated if this process involved the consideration of potential alternative interpretations or explanations (point 15 of the eMERGe Reporting Guidance). The synthesis authors say that they worked in a safe team environment in which they were able to disagree with each other: if there were disagreements about alternative interpretations, these might be usefully reported. The new conceptual framework is clearly set out in figure 2 and described in the section entitled 'Line of argument' on page 31. The main findings are summarised and compared to the existing literature in the Discussion section. Some of the limitations of the synthesis are described but perhaps more could be said about reflexivity and the impact of the research team on the synthesis

	findings. The recommendations are succinctly described, although they provide no guidance for clinicians who manage patients with chronic pain and have to sanction their return to work. Overall, the synthesis is reasonably well reported. As the eMERGe Reporting Guidance has only just been published, it may be unfair to expect the synthesis authors to address all 19 points in the Guidance. The reporting guidelines are not themselves directly concerned with the quality of the synthesis, although a reasonably well reported synthesis such as this one is likely to have been conducted well. In terms of quality, the synthesis does seem to have been well conducted. However I think that the synthesis authors could have gone further in their analysis. In particular, I would have liked to see a more systematic comparison of all 16 conceptual categories which might have provided explanations of the differences between upward and downward spirals, referred to in the last point of the line of argument. Are there ways in which the different conceptual categories work together to enhance return to work, or to inhibit it? Could separate lines of argument be formulated for upward and downward spirals, which might help clinicians in the same way that Malpass et al's (2009) 'decisive moral junctures' provide insights for those managing people taking antidepressants? The synthesis authors worked with a patient and public representative to develop their funding proposal, but this person seems not to have had any involvement with the synthesis itself or in the writing of the paper. Was such a contribution considered or discussed with the individual involved? I cannot follow the comment about CERQual made on page 34 and possibly other readers would not be able to follow it either. References A Booth. Searching for qualitative research for inclusion in systematic reviews: a structured methodological review. Systematic Reviews, 2016, 5(1), 74. N Britten, R Campbell, C Pope, J Donovan, M Morgan, R Pill (2002). Using meta ethnography to synthesise qualitative research: a worked example. Journal of Health Services Research and Policy; 7: 209-215. A Malpass, A Shaw, D Sharp, F Walter, G Feder, M Ridd, D Kessler. "Medication career" or "Moral career"? The two sides of managing antidepressants: a meta-ethnography of patients' experience of antidepressants. Social Science and Medicine, 2009, 68, 154-168.
--	--

VERSION 1 – AUTHOR RESPONSE

Reviewer 1

Comment

The authors state that the search terms used "included...", it would be helpful if the search strategy could be included either in a supplementary file or a note added to the text that the strategy is available on request.

Response

We have added the search strategy as Supplementary File 1.

Comment

The analysis of papers was principally carried out by one author, with two others checking the first ten papers (alphabetically). It is unclear whether this was a percentage of the papers identified or the rationale for selecting just 10 papers, could the authors justify this approach please?

Response

We have added justification for our approach (lines 134-6).

Comment

It would also be helpful to have a sense of how much agreement there was in identifying the third order concepts, it is reported that this took place over three meetings but was there any overlap in the concepts initially identified?

Response

This is addressed in lines 129-130.

Comment

The CASP tool was used to appraise the included papers, it is noted that all included studies passed the first two screening questions, were papers excluded if they did not pass these screening questions?

Response

This sentence has been rephrased to clarify meaning (lines 164-5).

Comment

There is a short section on the implications of the paper, I feel that this section would benefit from some further consideration. The authors suggest developing a return to work intervention to explore ways of addressing and managing chronic pain, managing work relationships and making workplace adjustments, but who would deliver such an intervention? Should it be located in healthcare, with the employer, another agency, or a combination? What are the likely implications of such an intervention on healthcare services and employers (improving health/changing workplace cultures)?

Response

This section has been developed and lengthened to answer the above questions (lines 372-385).

Reviewer 2

Comment

Line 136 - not sure if identifying the 11th version is necessary or relevant. You have identified that there were several revisions and I believe that is adequate.

Response

We have deleted the words "(the 11th version) which"(now line 146).

Comment

Line 136-139 - the sentence describing the methodology used by Toye et al doesn't read well, even just removing the 'and this' would help.

Response

We have deleted the words "and this" (now line 148).

Comment

Line 144 - how was the focus endorsed by the representative, when reading that it makes me want to have more information.

Response

This sentence has been edited to provide more information (now line 157).

Comment

Table 2 - I really liked this table as it clearly shows the relevance of the conceptual categories within the supporting studies. Although you have put quotations from the primary studies in this table I find that they are notably missing from the results section. I think incorporating different examples of quotes in the key conceptual categories descriptions would add more depth to the concepts.

Response

Additional quotations have been included in the descriptions to add more depth (lines 198 – 200, 203-205, 223-226, 230-232, 242-244, and 255-256).

Comment

Starting line 168 - There needs to be more information given here on the relevance of the three key conceptual categories as this is missing. You have a great explanation of them in lines 260-266 at the beginning of the discussion but I think a bit more is needed earlier. This paragraph from the discussion would even fit well in this first section of the overarching conceptual categories to give the reader a good explanation of the relevance and importance of the three key conceptual categories.

Response

We have moved lines 260-266 into results section (now lines 187-92).

Figure 2 – Conceptual model - This is a good model but it does seem unnatural going anti-clockwise and then this needs to be identified as you have done as anti-clockwise.

Response

An anticlockwise reading of the model was chosen so that concepts flowed, ending in a left to right direction. So we would like to keep it clearly identified as such, as we have done.

Comment

Line 303 – instead of saying 'carefully re-reading abstracts and then full texts' it might have more consistency if you said for example 'by applying the seven steps of the meta-ethnography process'.

Response

We agree this sentence is unclear so we have deleted it (now lines 346-7) as steps of meta-ethnography process are outlined elsewhere.

Comment

Line 340-341- the sentence on the influence of health and pain is overly wordy and confusing, but this isn't my subject area of expertise so it might be correctly pitched.

Response

We have discussed this sentence and we feel it is clear and we would prefer to keep it (now lines 403-4).

Reviewer 3

Comment

In particular the synthesis authors cite knowledge gaps in relation to return to work by people with chronic non-malignant pain, and the perspective of employers. However readers of BMJ Open may also be interested in the perspectives of doctors: perhaps the synthesis authors could at least indicate if there is any qualitative literature on this topic.

Response

A sentence has been added to introduction referring to qualitative research on this topic (lines 64-5).

Comment

In relation to the aim of the synthesis, the synthesis authors have gone beyond 'understanding obstacles' (abstract, page 2) and have provided a conceptual model of return to work. Thus it might be appropriate to revise the aim of the synthesis to reflect its more ambitious outcome (e.g. 'to provide a conceptual model').

Response

The aim has been revised as suggested in abstract (line 19) and in paper (line 75).

Comment

The synthesis authors list the databases searched and the search terms used but they do not describe the rationale for their search strategy (for example, why these databases were considered the most appropriate ones to use). They seem not to have used a search strategy specifically developed for qualitative studies (as discussed by Booth 2016 for example), which might perhaps have identified other qualitative papers.

Response

The text has been edited to describe the rationale (lines 97-9, 104-5).

Comment

They do not say who conducted the screening and selection of papers although possibly these were the same people who did the searching.

Response

The text has been edited to specify (lines 100-1) who did the searching.

Comment

Some of the exclusion criteria are given in the results section.

Response

This sentence has been rephrased (line 164) to clarify it is results.

Comment

The first column in table 2 seems to contain the synthesis authors' summaries (often referred to as third order concepts) and the second column contains participants' quotes (first order concepts). Thus it is not clear if the synthesis authors have worked with second order concepts.

Response

We did work with second order concepts and this is clarified by editing text (lines 184-6). Also punctuation in column of Table 2 has been changed to differentiate second and first order data and the latter are in now in italics.

Comment

The characteristics of the included studies, and the data extracted from each study, are clearly set out in Table 1. However there is no discussion of which aspects of the included studies were compared, nor how the studies were compared. Noblit and Hare (1988), whose approach the synthesis authors cite, refer to reciprocal and refutational syntheses, but it is not clear that the synthesis authors considered the methods of translation in these terms.

Response

Text has been added to clarify what we did (lines 140-3).

Comment

The synthesis authors describe the ways in which they developed overarching concepts and in particular how three of them independently developed their own conceptual models before comparing them and eventually agreeing a final (11th) version. It is not stated if this process involved the consideration of potential alternative interpretations or explanations (point 15 of the eMERGe Reporting Guidance).

Response

Text has been added to clarify and illustrate this process (lines 129-30).

Comment

The synthesis authors say that they worked in a safe team environment in which they were able to disagree with each other: if there were disagreements about alternative interpretations, these might be usefully reported.

Response

This was clarified in response to a comment from Reviewer 1 (lines 150-3).

Comment

Some of the limitations of the synthesis are described but perhaps more could be said about reflexivity and the impact of the research team on the synthesis findings.

Response

Text has been added to provide further explanation (lines 393-396)

Comment

The recommendations are succinctly described, although they provide no guidance for clinicians who manage patients with chronic pain and have to sanction their return to work.

Response

Text added in response to a similar comment from Reviewer 1 (lines 372-385).

Comment

As the eMERGE Reporting Guidance has only just been published, It may be unfair to expect the synthesis authors to address all 19 points in the Guidance.

Response

The importance of the eMERGE Reporting Guidance going forwards is acknowledged (lines 397-9).

Comment

However I think that the synthesis authors could have gone further in their analysis. In particular, I would have liked to see a more systematic comparison of all 16 conceptual categories which might have provided explanations of the differences between upward and downward spirals, referred to in the last point of the line of argument. Are there ways in which the different conceptual categories work together to enhance return to work, or to inhibit it? Could separate lines of argument be formulated for upward and downward spirals, which might help clinicians in the same way that Malpass et al's (2009) 'decisive moral junctures' provide insights for those managing people taking antidepressants?

Response

We have reflected on and discussed this and feel the line of argument and conceptual model demonstrate the extremely complex interaction between conceptual categories. It demonstrates that people with chronic pain need to be understood, believed, not judged and expectations need to be managed between employees and employers. It also shows that support can be given to manage pain, relationships and make work place adjustments, negotiate obstacles and navigate change but sometimes the person's thinking (health and pain representations), their level of self-belief, what work means to them, family expectations or system factors (e.g. type of job) may mean that even with this support it is not possible to return to work. Our additional section from lines 372 highlights the way in which it is highly individual how different factors can work together to enhance or inhibit return to work, and how these factors need to be assessed by the clinician working with the person with pain and the employer. Having thought about this carefully, for the above reasons we would like to keep our conceptual model diagram as it is.

Comment

The synthesis authors worked with a patient and public representative to develop their funding proposal, but this person seems not to have had any involvement with the synthesis itself or in the writing of the paper. Was such a contribution considered or discussed with the individual involved?

Response

We didn't involve the PPI representative in the synthesis, but would want to do this in future work.

Comment

I cannot follow the comment about CERQual made on page 34 and possibly other readers would not be able to follow it either.

Response

We have decided to delete this comment (lines 364-6).

VERSION 2 – REVIEW

REVIEWER	Gwenllian Wynne-Jones Arthritis Research UK Primary Care Centre, Keele University
REVIEW RETURNED	13-Mar-2019

GENERAL COMMENTS	I have enjoyed re-reading this paper, the authors have taken on board, and given full consideration, all the comments received making changes where necessary.
--

REVIEWER	Hazel Keedle, Lecturer Western Sydney University Australia
REVIEW RETURNED	28-Feb-2019

GENERAL COMMENTS	I think the changes you have made to the manuscript have enhanced the paper and that it is now ready for publication.
---